# Shell colour luminance of Cuban painted snails, *Polymita picta* and *Polymita muscarum* (Gastropoda: Cepolidae)

**Mario Juan Gordillo-Pérez**[1]*, **Natalie Beenaerts**[1], **Dunia L. Sánchez**[2], **Karen Smeets**[1], **Yaumel Calixto Arias-Sosa**[3], **Bernardo Reyes-Tur**[4]

1 Centre for Environmental Sciences, Hasselt University, Hasselt, Belgium, 2 Facultad de Ciencias Técnicas y Agropecuarias, Departamento de Agronomía, Universidad de Las Tunas, Las Tunas, Cuba, 3 Facultad de Ciencias Naturales y Exactas, Departamento de Física, Universidad de Oriente, Santiago de Cuba, Cuba, 4 Facultad de Ciencias Naturales y Exactas, Departamento de Biología y Geografía, Universidad de Oriente, Santiago de Cuba, Cuba

* mariojg755@gmail.com

**Data Availability Statement:** All relevant data are within the manuscript and its Supporting Information files.

## Abstract

Climate change is a global environmental threat, directly affecting biodiversity. Terrestrial gastropods are particularly susceptible to alterations in temperature and humidity and have develop morph-physiological and behavioural adaptations in this regard. Shell colour polymorphism and its potential implication for thermoresistance constitute an unexplored field in Neotropical land snails. The variation in shell colour luminance is characterized in the threatened endemic Eastern Cuban tree snails *Polymita picta* and *Polymita muscarum* using digital tools; being able to discriminate shell luminance between colour morphs for both species, under different image-taking conditions. For *P. muscarum*, the albino morph presented the highest luminance values (152.7 ± 0.4); while the lowest values correspond to the brown morph with dark bands (112.9 ± 0.8). Otherwise, for *P. picta*, the morphs showing the highest luminance were yellow with a pink sutural band (112.8 ± 7.1) and pale yellow (112.6 ± 10.3) and the lowest luminance corresponded to the black morph (44.5 ± 1.2). The presence of dark bands decreased the luminance values regardless of their position in the shell, the morph and the species analysed. In general, the shells of *P. muscarum* have higher luminance than those of *P. picta*. Luminance variations demonstrate the 'indiscrete' nature of this trait and highlight the complex interactions between evolutionary mechanisms and shell color polymorphism in *Polymita*. This supports the hypothesis that colour has adaptive value for thermoregulation, encompassing not only the background colour but also the coloration of the bands. The differences in the shell luminance in both species suggest a correlation with the geographical distribution and corresponding habitats. Based on our findings, yellowish morphs will be more resistant to future climatic conditions in their respective habitats on the island.

**Funding:** The author(s) received no specific funding for this work.

**Competing interests:** The authors have declared that no competing interests exist.

## Introduction

Climate change is considered as one of the global threats to biodiversity affecting physical traits, abundance and distribution of many species' groups [1,2]. Some land snail species appear to be especially susceptible to climate change [3,4]. Their limited dispersal ability hinders timely colonization of new habitats with favourable environmental conditions [5]. Wildlife in the oceanic and Neotropical Caribbean islands even appears to be more susceptible to climate change [6–8]. Among the West Indies fauna, Cuban land snails exhibit a high level of species endemism and are mainly occurring at the mountainous regions [9,10]. Even minor changes in climate conditions in these regions could have significant impact on the ecological niche of many species, potentially leading to population reductions and even extinction [11–13].

Elevated temperatures and decreasing humidity are known threats for land snails [5,14,15]. These environmental stressors affect their physiology by disrupting their normal water balance, leading to desiccation and potential death [16–18].

Terrestrial gastropods possess behavioural, physiological, and morphological adaptations to persist in a different climatological environment [19,20]. Shell traits, such as size, pigmentation, texture and aperture diameter, play a role in the ability of land snails to withstand contemporary harsh climate in their current habitats [21]. Additionally, the interaction of these shell traits directly influences the soft body temperature and water balance of land snails [22], leading to research focusing on thermoregulation [23–25]. However, to the best of our knowledge, most research addressing the above issues has predominantly focused on non-Neotropical gastropods [4,18,19,21,26].

Currently the European snail *Cepaea nemoralis* (Linnaeus, 1758) is the most pre-eminent model of shell colour polymorphism. This species exhibits three perceptual base colours and a complex band pattern controlled by a 'supergene', whereby multiple linked loci result in distinct phenotypes [27,28]. Essentially, the *Cepaea* model has provided evidence of the modulation of this polymorphism through the influence of natural selection in natural populations [27,29–31] However, recently, ground-breaking publications on the species have challenged the 'Fordian theory of polymorphism' [32] and the traditional view that 'supergenes control discrete complex phenotypes' [33,34]. This 'indiscrete' nature of the *C. nemoralis* polymorphism indicates that chromatic variation in shells is continuously distributed within visual space.

In the Neotropical region, the emblematic genus *Polymita* Beck (1837) comprises six endemic, critically endangered species occurring in Eastern Cuba [35–38]. To further halt their decline *Polymita* species are now protected in Cuba and listed under CITES (Convention on International Trade in Endangered Species of Fauna and Flora) Appendix II [39]. These tree snails, worldwide admired for their beauty, exhibit a vivid polymorphism of shell colour and banding patterns [40,41].

The *Polymita* genus, its species, subspecies, and varieties were comprehensively summarized in the work *The Genus Polymita* by Don Carlos de la Torre y Huerta [38]. However, the subspecific categories require revision and verification through molecular genetics, as well as new ecological and behavioural studies. Since the 1980s, several authors have focused on the genetic polymorphism and ecology of *Polymita* spp. Shell colour and banding patterns were consistently treated as discrete Mendelian inherited traits, despite suggestions that epistasis, pleiotropy, or environmental influences could also play a role [42–50].

The species *Polymita muscarum* Lea (1834) inhabits a range of environments, including coastal vegetation, more humid forests, and even highly anthropized ecosystems in the northern regions of Eastern and Central-Eastern Cuba [36–38]. The shells of *P. muscarum* display

two background colours, white and brown [42]. By combining these base colours with or without spiral bands, and variations in band number and tone, six distinct phenotypes can be identified [42,47]. In contrast, *Polymita picta* Born (1780) is restricted to coastal xerophytic vegetation, rainforests, and agroecosystems in the municipalities of Baracoa and Maisí, located in the North-Eastern Cuban region [35]. The shells of *P. picta* exhibit four primary background colours—white, yellow, red, and brown—that combine to form ten different phenotypes [38,47]. In general, the observation of polymorphism in specific habitats is not considered to be coincidental but rather an evolutionary adaptation to the environment [42–45].

Contrary to the *Cepaea* model, *Polymita* polymorphism cannot straightforwardly be explained through natural selection [45]. Moreover, the previously employed qualitative methodologies to classify colour in *Polymita* [49] may introduce bias, highlighting the need to develop quantitative procedures for colour description [44].

The International Commission on Illumination (CIE) has established the human colorimetric system, based on the stimulation of the different photoreceptor cells in the retina [51]. Three representative parameters have been identified for each cone cell, i.e. tri-stimulus values (XYZ). This system refers to the primary colours (red, green and blue) providing a standard reference for other colour spaces, including RGB. According to CIE, in the XYZ colour system, the Y coordinate corresponds to the luminance value, which is defined as the luminous efficiency of the human eye across the entire visible spectrum [52,53]. In digital images, each colour component includes 256 luminance levels in eight bits, where 0 represents the black and 255 represents the white. Then, the colour depth or bits per pixel represents the range of colour that can be shown in an image [54].

In this study, we aim to characterize the shell colour luminance in *P. picta* and *P. muscarum*. Additionally, we also explore its potential relationship with thermoregulation. To our knowledge, we are the first to perform a quantitative approach to reveal the shell chromatic properties in Caribbean land snails. Consequently, this research establishes the methodological baseline for a future quantitative analysis to enable elucidating the relationship between colour polymorphism, luminance and thermoresistance.

## Materials and methods

### Samples and phenotype classification

In this study we included adult individual shells of *P. muscarum* (N = 180) and *P. picta* (N = 77). The analysed shells of *P. muscarum* belong to the collection of the Natural History Museum Charles T. Ramsden de la Torre at Universidad de Oriente, Santiago de Cuba, Cuba (find repository information in S1 Table). Each shell was classified into one of the six phenotypic groups according to shell background and spiral band colours: white with dark bands, white without bands, brown with dark bands, brown with white bands, brown without bands and albino [42]. The shells of *P. picta* belong to the collection of the Terrestrial Mollusk Captive Breeding Laboratory, Universidad de Oriente, Santiago de Cuba (find repository information in S4 Table). Shells were classified into five phenotypic groups according to their background colours: black, red-orange, green, brown and yellow. Yellow shells were subdivided considering the colour of the sutural band (pink or dark) and the background tonality. For this species we followed the shell character denomination criteria of Berovides *et al.* [48]. No permits were required for the described study, which complied with all relevant regulations.

## Image processing and colour densitometry

As previously reported for various *Polymita* species, individuals frequently remain in a vertical or horizontal, on upside-down position, often utilizing branches and stems during diurnal rest and long lasting estivation [55–58]. During these physiological states, the snails attach to surfaces by a delicate, transparent veil or a hard epiphragm, requiring sufficient contact area that typically covers the ventral surface. Therefore, the shell dorsal surface is more directly in contact with sunlight than the ventral surface. Considering these factors, we opted to photograph the shells from a dorsal view. Initially, we tested for statistically significant differences in luminance between the dorsal and ventral views (S1 Table). Each shell was fixed on a vertical surface with a white background and illuminated with an intensity of 1066 ± 1.5 lx using a fluorescent lamp. Photos were taken from a fixed distance of 15 cm using a digital camera Nikon-Coolpix s9600 in macro mode (always with flash). The colour densitometry was carried out using the software ImageJ 1.52a in Java 1.8.0_112 [59]. Subsequently the images were decomposed in their red (R), green (G) and blue (B) components using the *Split Channels* function in the colour menu. The photos' frame was cropped to the borders of the shells using the tool *Rectangle*, diminishing the effect of the white background on the luminance measuring. A histogram for each colour component was obtained and the RGB values were measured using the function *Histogram*. The representative luminance value Y, in scale 0 (black) to 255 (white) was calculated using the formula $Y = 0.299R + 0.587G + 0.114B$ [53,60] (S1 Fig).

## Shell colour luminance: Preliminary analysis using different light conditions

Shells representing three morphs of *P. muscarum* (white banded brown, unbanded brown and unbanded white) and two of *P. picta* (banded brown and pale yellow) were randomly selected (N = 6 for each group). Three conditions for picture taking were tested (i.e. indoor natural light at 9:00 hrs corresponding to 261 ± 6 lx; indoor natural light at 14:00 hrs corresponding to 397 ± 31 lx and artificial light corresponding to 1066 ± 1.5 lx. Pictures were taken in the Terrestrial Mollusk Captive Breeding Laboratory, Universidad de Oriente, Santiago de Cuba on April, 2022.

## Shell colour luminance

For *P. muscarum* shell colour luminance variation between the six phenotypic groups was compared (N = 30 for each group). For *P. picta* this comparison was conducted between the five phenotypic groups: black, red-orange and green shells (each morph N = 10); brown (N = 19) and yellow (N = 28). Moreover, we subdivided the yellow morph of *P. picta* again into three morphological types: dark banded yellow, pink banded yellow and pale yellow based on the variation in shell background and sutural band colours.

## Statistical analysis

We performed a Shapiro-Wilk test to analyze the normality in the data distribution and a Levene test for homoscedasticity on our data. In the preliminary analysis using different light conditions, we conducted a repeated measure ANOVA test for the groups with a parametric data distribution; for the group with a non-parametric data distribution we chose a Friedman test. Considering the limited sample size, and limited numbers of significant statistical differences, we carried a Wilcoxon signed rank test with Bonferroni corrections out and an $\eta^2$ (eta squared) for effect size.

For the studies of shell colour luminance in *P. muscarum* and *P. picta*, and the luminance variation in yellow *P. picta* morphs, and we carried out an ANOVA test to check for normal distributed data. We applied the non-parametric Kruskal-Wallis test on the other datasets. However on the datasets with significant differences we performed Tukey and Dunn *post hoc* tests. All analysis, including descriptive statistics calculation and graphical outputs were performed using R software [61].

## Results

In four of the five phenotypic groups, no significant differences were detected in mean shell colour luminance across the three light conditions, despite the obvious consequence of the availability having of fewer samples per analysis and the consequent lower statistical power (Fig 1A–1E). However, for the unbanded brown shells of *P. muscarum* (Fig 1B), the luminance means were significantly different (Friedman chi-squared = 10.333, P = 0.005704). The luminance values for both natural light conditions were distinguishable when compared to artificial light (09:00 hrs vs. Artificial: P = 0.004998, Bonferroni-adjusted P = 0.014994; 14:00 hrs vs. Artificial: P = 0.004998, Bonferroni-adjusted P = 0.014994). Light conditions exert a moderate influence on the dependent variable ($\eta^2$ = 0.694), indicating that a substantial portion of the variability is explained by these factors. We supply all the experimental data in S2 Table. Following these results we decided to continue all our further analyses under artificial light conditions.

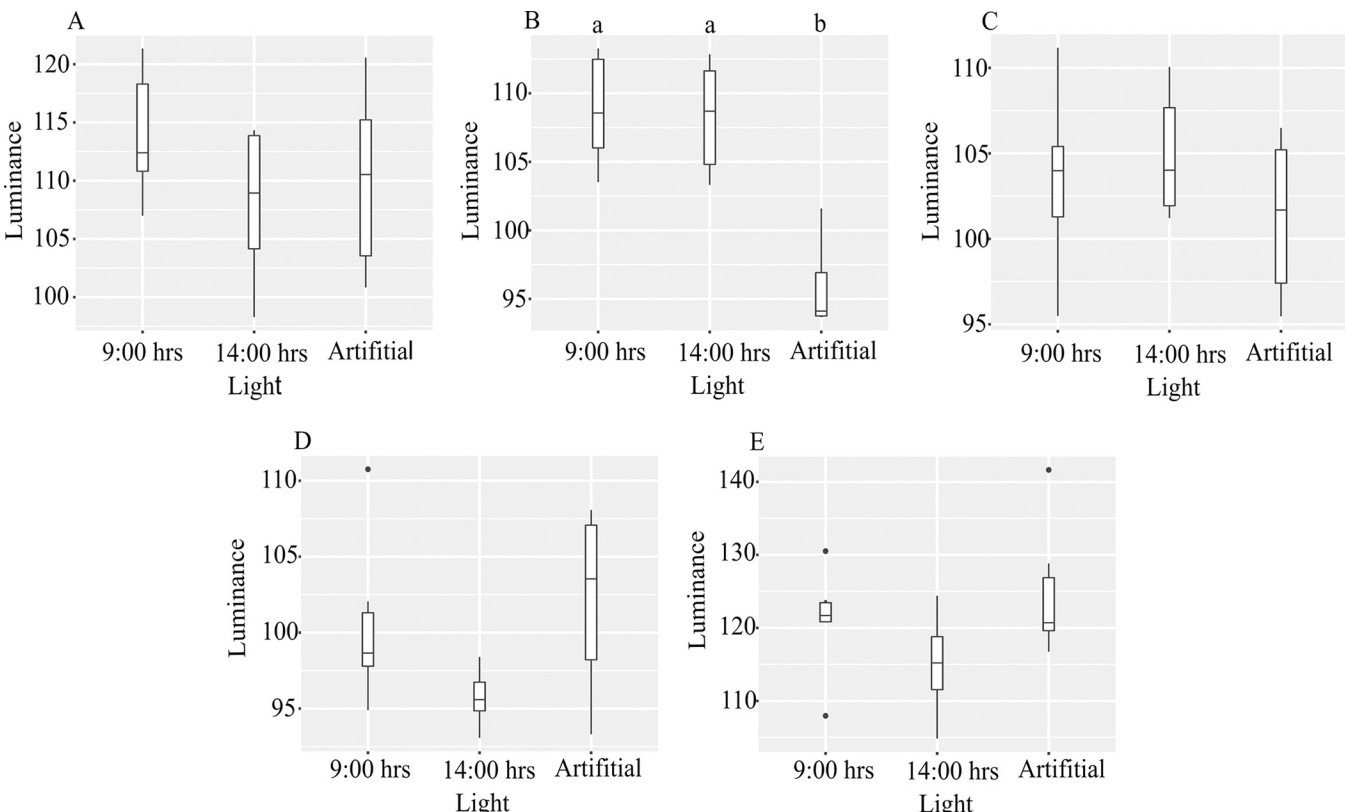

**Fig 1.** Shell colour luminance variation of *Polymita* according to background colour and spiral band categories at three light conditions (indoor natural light 9 AM, indoor natural light 2 PM and artificial light) for *P. muscarum* (A-C) and *P. picta* (D, E). A. white banded brown. B. unbanded brown. C. unbanded white. D. brown. E. pale yellow. Mean ± SE are represented in box plots. Different letters show significant differences between groups, P < 0.05. For each group N = 6.

## Shell colour luminance in *Polymita muscarum*

Light coloured shells, i.e. albino and unbanded white, had the higher luminance values, 143.6 ± 0.5 and 137.1 ± 1 respectively, while the darker morphs possess the lower values. Significant differences were observed between phenotypic groups (H = 120.42, P = 2.2e-16) and the *post hoc* test allowed to merge the morphs based on the statistical differences (a: dark banded brown; b: unbanded brown, white banded brown and dark banded white; c: unbanded white; and d: albino, respectively (Fig 2, S3 Table). The probability density diagrams indicate greater data dispersion in the unbanded and white-banded brown shell morphs, as well as in the dark-banded white, while the least dispersed data are found in the albino group. A multimodal distribution (at least two modes) is observed in the dark and white banded brown morphs and in albinos (Fig 2).

## Shell colour luminance in *Polymita picta*

The highest luminance values were found in shells with lighter colours, i.e. yellow, 107.3 ± 3 and brown, 95.1 ± 3; the lowest values were observed in black shells. Red-orange and the green morphs are located in the middle of the luminance spectrum. Significant differences between morphs were revealed (F = 20.1, P = 1.17e-12) and the *post hoc* test allowed to group the

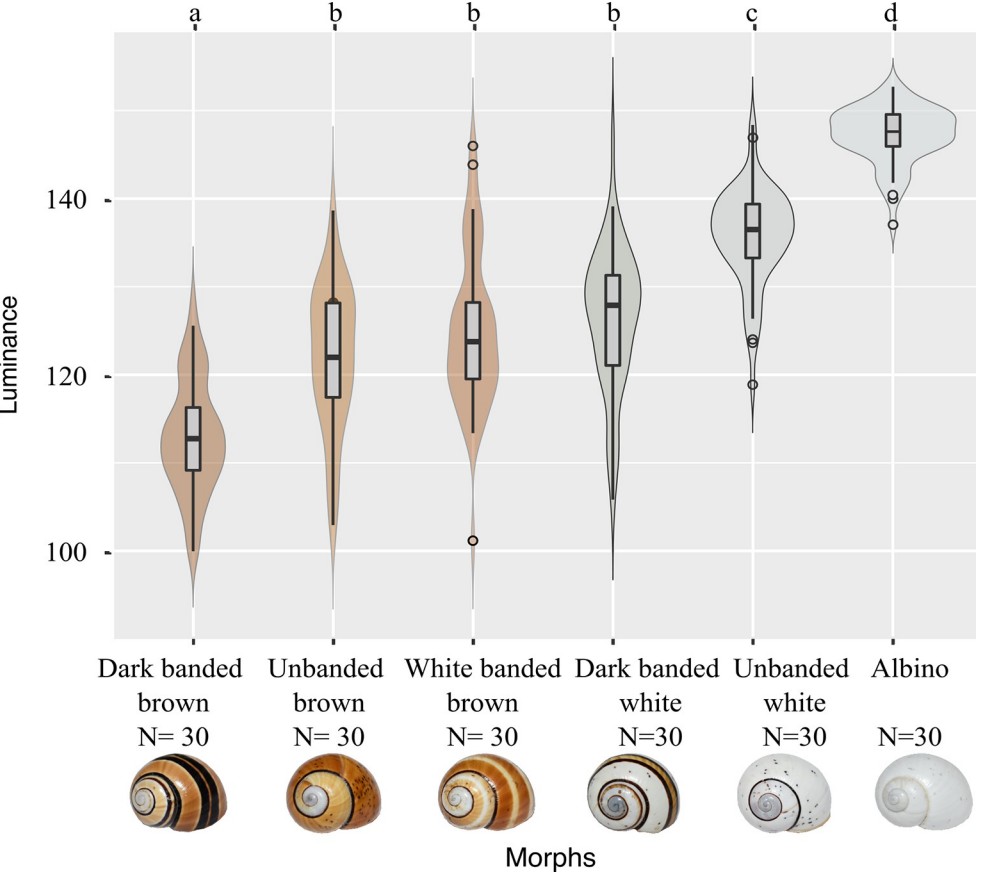

**Fig 2. Shell colour luminance variation of six phenotypic groups according to background colour and spiral band categories of *Polymita muscarum*.** Mean ± SE are represented in box plots and values frequencies in violin diagrams. The letters (a-d) on top correspond to significant differences between groups (a ≠ b ≠ c ≠ d) for p < 0.001. Luminance is represented on a scale from 0 (black) to 255 (white). For each group N = 30.

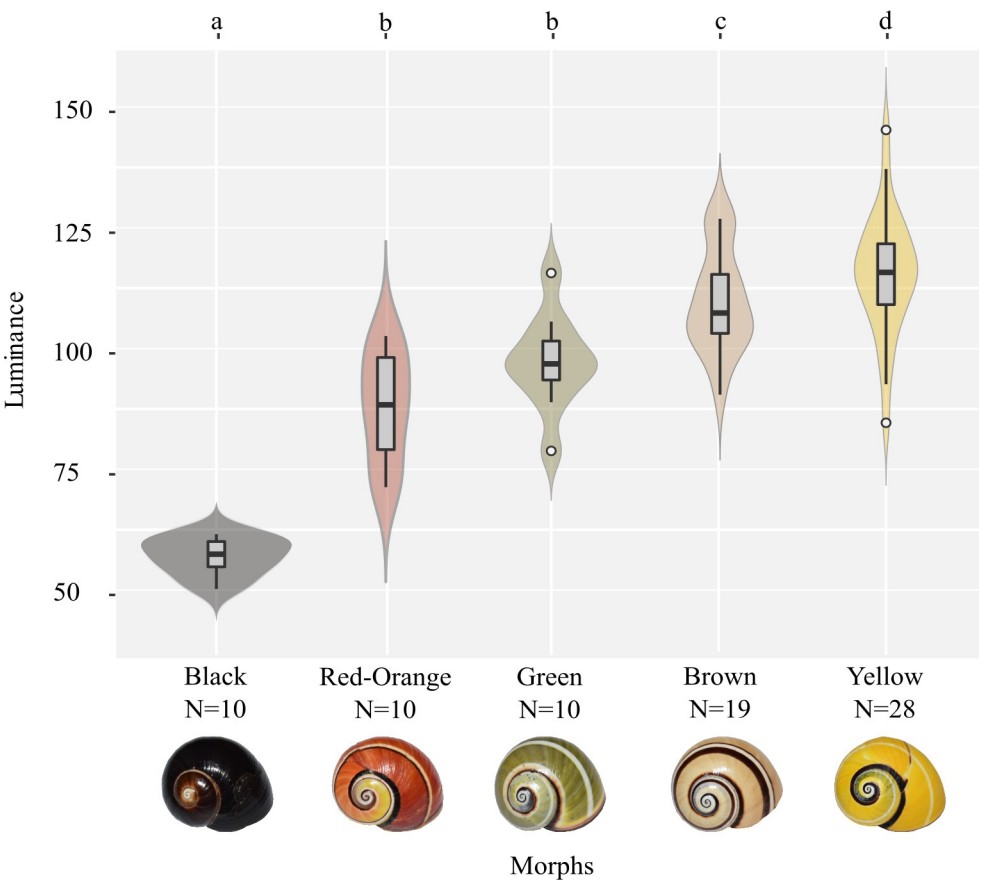

**Fig 3. Shell colour luminance variation of five phenotypic groups according to background colour categories of *Polymita picta*.** Mean ± SE are represented in box plots and frequencies in violin diagram. The letters (a-d) on top correspond to significant differences between groups (a ≠ b ≠ c ≠ d) for $p < 0.05$. Luminance is represented on a scale from 0 (black) to 255 (white).

morphs according to its differences (a: black; b: red-orange and green; c: brown; d: yellow) (Fig 3, S4 Table). We observe the greatest data dispersion in the yellow morph, while the least dispersion is found in the black one. The probability density diagram shows a multimodal distribution at least in the brown morph (Fig 3).

### Shell colour luminance: Variation of yellow shell morphs of *Polymita picta*

The analysis of the luminance between three variants of the yellow morph of *P. picta* indicates the highest luminance value for the pale yellow shells (112.6 ± 10.3), followed by the pink banded yellow morphs and the dark banded ones, respectively (Fig 4, S4 Table). The statistical analysis revealed significant differences between the groups, (F = 9.5, P = 8.1e-4) and the *post hoc* test allowed the grouping of morphs (a: dark banded yellow; b: pink banded yellow, pale yellow) (Fig 4).

### Discussion

Thermal resistance appears to be a crucial adaptation to cope with the effects of climate change in the Neotropics. More specifically, the Caribbean region, an acknowledged biodiversity hotspot, is under threat due its insular biogeography, the presence of extreme meteorological

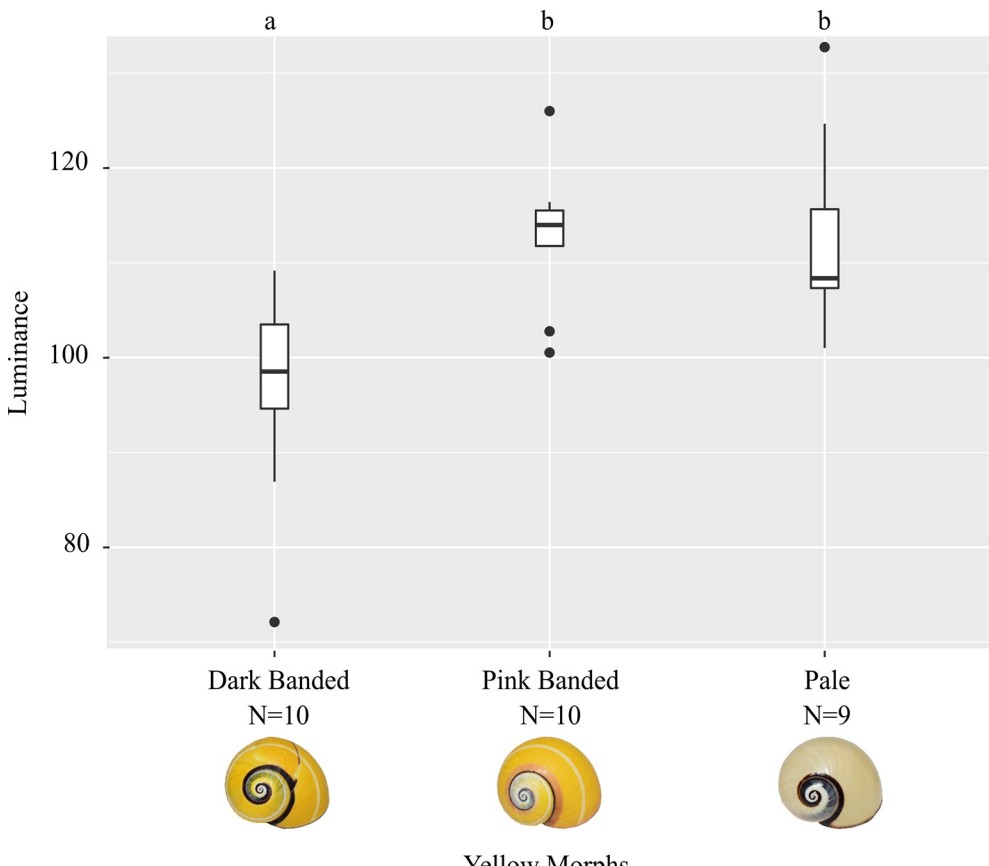

**Fig 4. Shell colour luminance variation of three phenotypic groups according to background and sutural band colour categories for yellow shells of *Polymita picta*.** Mean ± SE are represented in box plots and different letters showed significant differences between groups, P < 0.001. Luminance is represented on a scale from 0 (black) to 255 (white).

events, anthropogenic activities, sea level rise and global warming in general. To elucidate the potential role of shell colour in thermoregulation in Caribbean land snail species, we selected two conspicuous Cuban endemic tree snails. The distinct polymorphism of their shells, their well-known distribution and habitats make them ideal study species. Contrary to previous studies on *Polymita* polymorphism, we used a quantitative imaging methodology that allows for statistical discrimination between phenotypic groups based on the shell colour luminance. Additionally, we present a first objective assessment of the intra- and interspecific variation in shell colour luminance enabling the estimation of the proportion of visible reflected radiation, or albedo [25].

## Shell colour luminance in *Polymita muscarum* and *Polymita picta*

Luminance is a proper parameter to assess colour variation and indirectly albedo, therefore it allows us to relate shell polymorphism to thermoregulation. As expected, the lowest luminance was consistently recorded in the darker morphs (i.e. for *P. muscarum* the dark banded brown morph; for *P. picta* the black one) while the highest luminance was consistently observed in the lighter ones (i.e. for *P. muscarum* albino and unbanded white; for *P. picta* the pale yellow morph). Additionally, the unbanded brown shells of *P. muscarum* showed higher luminance

than the dark banded brown individuals but a lower luminance than the brown ones possessing white bands.

Previous studies using morphological characters of *Polymita* such as the band and background colours showed paler morphs occurred more frequently in very sunny locations, while the darker ones are more common in shady habitats [7,42,43]. Furthermore, humidity and temperature are considered as climatic drivers in the selection of *P. picta* [8,44]. Individuals with a white shell survive the highest temperatures if treated with humid hot air, but when combined with dry hot air, the yellow phenotype is more resistant [44].

Nevertheless, we expect that a future follow-up study including additional variables, such as shell texture, porosity and aperture size, that play a role in thermoregulation [19] could disentangle more distinct contributions within the complex relationship between luminance, reflected light and thermoresistance. We suggest solar radiation is an ecological factor driving climatic selection in both *Polymita* species, because shells with a high luminance value reflect the sun light more efficiently and consequently facilitate the necessary lower soft body temperature in a warm environment. In temperate climates animals with low shell luminance are favoured due to the higher solar radiation absorption allowing them to easily reach an optimal physiological temperature.

The data distribution exhibited two main modes within three morphological groups of *P. muscarum* (dark-banded brown, white-banded brown, and albino) and in at least one morphological group of *P. picta* (brown) (Fig 2); in the remaining cases, a unimodal distribution was observed. Given the presence of bimodal and unimodal distributions, the potential inclusion of individuals belonging to different populations, as well as the rather limited sample size of our study, we conclude that the genetic and evolutionary mechanisms interacting with colour polymorphism in *Polymita* warrant thorough investigation in the future. This would allow us to relate these distributions to earlier proposed mechanisms, such as the action of stabilizing selection in *Polymita picta roseolimbata* [50] or epistatic interactions in *P. muscarum* [47] or incomplete penetrance, such as shown in *C. nemoralis* [33,62–64].

It appears that the evolutionary mechanisms influencing colour polymorphism in *Polymita* differ from those reported for non-Neotropical species and previously suggested for the genus, thereby establishing it as a model species for studying these mechanisms in the Neotropics.

The variation in dispersion, particularly among morphs with intermediate luminance values in *P. muscarum* and related colours such as yellow, red-orange, and brown in *P. picta*, further supports the concept of 'continuity' in colour variables for both species. Through this first quantitative approach to chromatic variation in *Polymita*, we have been able to demonstrate the 'indiscrete' nature of base colour and banding patterns, as well as the presence of predominant morphological variations within single distinguishable morphs. These findings open new avenues for future research utilizing genetic and genomic approaches.

We propose that solar radiation is an ecological factor driving climatic selection in both *Polymita* species. Shells with higher luminance values reflect sunlight more efficiently, aiding in maintaining lower soft body temperatures in warm environments. In temperate climates, animals with lower shell luminance are favoured, as their higher absorption of solar radiation allows them to more easily reach optimal physiological temperatures [25,65–69]. According to us, the quantitative assessment of *Polymita* shell luminance provides more objective evidence regarding the importance of different environmental traits. Based on the results presented here, the quantitative assessment of *Polymita* shell luminance offers more objective evidence regarding the potential contribution of various environmental factors and provides insights into the evolutionary mechanisms influencing *Polymita* populations. This methodology allows to better understand the potential adaptive value of its shell colour polymorphism.

Our findings are consistent with inferences made for non-Neotropical land snails. A relationship between solar radiation, humidity, and thermoresistance in different morphs of at least four European species has been demonstrated, providing an explanation for microhabitat selection [4,25,26,66,70–72]. Unfortunately, in some of our shells, the exact locations are unknown and temporary monitoring of abundances is lacking; therefore, we fail to make inferences on microhabitats and further effects of weather changes due to climate change, land use changes or pollution.

Not investigated in this research, but possibly highly relevant would be the measurements of oxidative stress levels in different morphs because shell pigmentation appears to have an influence on survival in elevated temperatures [73–76]. Therefore, we could expect lower oxidative stress levels in morphs with higher luminance values.

Remarkably, the higher luminance values were founded in shell colour luminance of *P. muscarum*. was higher than for *P. picta*. Even once is not appropriate to statistically compare between the two species, this interspecific difference might be related to their different evolutionary histories and environmental habitat conditions [36]. Also, their behavioural traits, for example, their daily and yearly activity rhythms, corroborate this hypothesis [77,78]. In the case of *P. picta* have been reported both, unimodal and bimodal daily activity patterns in different seasonal periods [79], in contrast *P. muscarum* only exhibited an unimodal daily activity pattern [77]. Furthermore, *P. picta* inhabits the Cuban region with the highest precipitation levels and probably because of a lower exposition to stressful weather conditions, its mating is at least twice as long as in its congeners *P. venusta* and *P. muscarum* [7,36,50,80]. Otherwise, typically, populations of *P. muscarum* are occurring in areas with limited water availability and high irradiation levels, like xerophitic vegetation in coastal zones [38,80,81].

## Shell colour luminance: Variation in yellow shell morphs of *Polymita picta*

The sutural band colour (black or pink) in shells with the same background colour (yellow) significantly change the luminance values. Early taxonomists already recognized the relevance of the sutural band, i.e. individuals with a pink sutural band are described in the literature as subspecies *P. picta roseolimbata* [38,43,48] and the ones with a black sutural band are known as subspecies *P. picta nigrolimbata* [38]. We found that using a more solid statistical approach, the taxonomic view still seems to remain intact for these two subspecies. Notwithstanding, we recommend a thorough revision of the genus *Polymita*, including biogeographical, morphological and molecular insights.

We suggest that thermal selection is a key evolutionary mechanism driving the contemporary distinct distribution of both subspecies. The distinct inter- and intraspecific luminance variation in the shells in *P. muscarum* and *P. picta* support the hypothesis that background and band colour polymorphism have adaptive value in thermoregulation. Furthermore, the interspecific differences in shell colour luminance suggest a relationship with their contemporary geographical distribution and habitat.

This study, together with similar findings from non-Neotropical regions and in conjunction with environmental warming projections for Cuba and other Caribbean islands [82], suggests a potential future decrease in phenotypic variability in *P. muscarum* and *P. picta*, with a likely dominance of yellowish morphs. In a follow-up study an initial exploration of the potential relation between shell colour luminance and shell pigments, as well as the cellular and molecular responses to thermal stress in *Polymita*, could contribute to unravel even more the potential effect of climate change on these colourful snails and consequently improve the design of conservation strategies and actions.

## Conclusion

This study presents the first quantitative approach to the chromatic characteristics of *Polymita* using digital imaging. The measured luminance allows for statistical discrimination of morphs and highlights the 'indiscrete' nature of this trait, contributing theoretically to the understanding of polymorphism. The distinct polymorphism of shell colour in *Polymita muscarum* and *Polymita picta* is critical for thermoregulation, with lighter morphs exhibiting higher luminance, suggesting more efficient reflection of solar radiation. Ecological factors such as solar radiation and habitat conditions emerge as key drivers of selection in these species. However, the distribution of luminance data reveals both unimodal and multimodal patterns, indicating the complexity of the evolutionary mechanisms influencing *Polymita* polymorphism. This suggests that the evolutionary mechanisms affecting colour polymorphism in these Neotropical species may differ from those observed in non-Neotropical counterparts, establishing *Polymita* as a model for future research on adaptive evolution in Neotropical ecosystems.

Furthermore, the observed differences in luminance among yellow individuals may reflect their unique evolutionary histories and ecological adaptations. The findings regarding yellow morphs of *P. picta* emphasize the need for a reevaluation of taxonomy within the genus, suggesting a correlation between shell colour variation and habitat distribution. Given the projected effects of climate change on the Caribbean, we anticipate a potential reduction in phenotypic variability among *Polymita* species, with lighter morphs likely becoming more prevalent. Future research should investigate the relationships between shell colour, thermal stress, and molecular responses to better inform conservation strategies for these endemic species in the face of environmental changes.

## Supporting information

**S1 Table. Luminance of dorsal and ventral views in *Polymita muscarum*.**
(XLS)

**S2 Table. RGB components and luminance of the morphs used for light experiments.**
(XLS)

**S3 Table. RGB components and luminance of *Polymita muscarum* shells.**
(XLS)

**S4 Table. RGB components and luminance of *Polymita picta* shells.**
(XLS)

**S1 Fig. Image processing diagram.**
(TIF)

## Acknowledgments

We greatly appreciate suggestions, language corrections and writing assistance of Angus Davison, editor and referees. We would also like to thank to the Natural History Museum Charles T. Ramsden de la Torre at Universidad de Oriente (Cuba), VLIR-UOS Project: Valorization of Eastern Cuban biodiversity in a climate change scenario (Cuba-Belgium), NABU (Germany) and EXOCOM, Tim Claesen (Belgium).

## Author Contributions

**Conceptualization:** Mario Juan Gordillo-Pérez, Natalie Beenaerts, Dunia L. Sánchez, Yaumel Calixto Arias-Sosa, Bernardo Reyes-Tur.

**Data curation:** Mario Juan Gordillo-Pérez, Dunia L. Sánchez, Bernardo Reyes-Tur.

**Formal analysis:** Mario Juan Gordillo-Pérez, Natalie Beenaerts, Dunia L. Sánchez, Karen Smeets, Yaumel Calixto Arias-Sosa, Bernardo Reyes-Tur.

**Investigation:** Mario Juan Gordillo-Pérez, Dunia L. Sánchez, Yaumel Calixto Arias-Sosa, Bernardo Reyes-Tur.

**Methodology:** Mario Juan Gordillo-Pérez, Dunia L. Sánchez, Karen Smeets, Yaumel Calixto Arias-Sosa, Bernardo Reyes-Tur.

**Resources:** Mario Juan Gordillo-Pérez.

**Software:** Mario Juan Gordillo-Pérez, Dunia L. Sánchez.

**Supervision:** Mario Juan Gordillo-Pérez, Natalie Beenaerts, Karen Smeets, Yaumel Calixto Arias-Sosa, Bernardo Reyes-Tur.

**Validation:** Mario Juan Gordillo-Pérez, Natalie Beenaerts, Dunia L. Sánchez, Karen Smeets, Yaumel Calixto Arias-Sosa, Bernardo Reyes-Tur.

**Visualization:** Mario Juan Gordillo-Pérez, Natalie Beenaerts, Bernardo Reyes-Tur.

**Writing – original draft:** Mario Juan Gordillo-Pérez, Natalie Beenaerts, Dunia L. Sánchez, Yaumel Calixto Arias-Sosa, Bernardo Reyes-Tur.

**Writing – review & editing:** Mario Juan Gordillo-Pérez, Natalie Beenaerts, Karen Smeets, Yaumel Calixto Arias-Sosa, Bernardo Reyes-Tur.

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
