## [Decision Letter · Decision Letter 0]

8 Aug 2024

PONE-D-24-25732Shell colour luminance of Cuban painted snails, Polymita picta and Polymita muscarum (Gastropoda: Cepolidae)PLOS ONE

Dear Dr. Gordillo Pérez,

Thank you for submitting your manuscript to PLOS ONE. After careful consideration, we feel that it has merit but does not fully meet PLOS ONE’s publication criteria as it currently stands. Therefore, we invite you to submit a revised version of the manuscript that addresses the points raised during the review process.

We look forward to receiving your revised manuscript.

Kind regards,

Hudson Alves Pinto, Ph.D

Academic Editor

PLOS ONE

2. In your manuscript, please provide additional information regarding the specimens used in your study. Ensure that you have reported human remain specimen numbers and complete repository information, including museum name and geographic location. 

For more information on PLOS ONE's requirements for paleontology and archeology research, see https://journals.plos.org/plosone/s/submission-guidelines#loc-paleontology-and-archaeology-research.

Additional Editor Comments:

Dear Authors,

I invite the authors to revise the MS, taken in count especially the comments presented by Reviewer 1.

I look forward to receiving a new version of this interesting work.

Best regards,

Hudson

Reviewers' comments:

Reviewer's Responses to Questions

**Comments to the Author**

1. Is the manuscript technically sound, and do the data support the conclusions?

Reviewer #1: No

Reviewer #2: Yes

Reviewer #3: Yes

2. Has the statistical analysis been performed appropriately and rigorously? 

Reviewer #1: No

Reviewer #2: Yes

Reviewer #3: Yes

3. Have the authors made all data underlying the findings in their manuscript fully available?

Reviewer #1: No

Reviewer #2: Yes

Reviewer #3: No

4. Is the manuscript presented in an intelligible fashion and written in standard English?

Reviewer #1: Yes

Reviewer #2: Yes

Reviewer #3: Yes

5. Review Comments to the Author

Reviewer #1: The manuscript PONE-D-24-25732 aims to corroborate a basic prediction of evolutionary ecological explanations for the presence of shell color polymorphism in Polymita species from Cuba and other regions. Specifically, it suggests that color is influenced and maintained by the varying sunny environments in different populations and microhabitats. The study uses approximately 180 specimens from one species and about 150 from another, sourced from a museum collection. The authors photographed these specimens and used software to analyze the colorimetry and luminance value based on a methodology by Schilk (1995). The manuscript, however, lacks substantial scientific value, because study something obvious which do not add anything new to the system (see below) and uses very little experimental effort (see below). The authors should revise their objectives and methodology to meet international standards before attempting publication. Here are some general comments that may help the authors in the future:

Literature Review. Many classical references on Polymita (in Spanish) are of low scientific quality or are supposed to proof what they really proof. It might be beneficial to prepare an English review that objectively summarizes previous work and realistically presents the data. Currently, there is no clear proof that Polymita polymorphism is maintained by natural selection, despite some indirect suggestions. A comprehensive review could set a solid foundation for future research.

Use of Luminance. The methodology for using luminance as presented has little empirical value, especially with museum specimens that have been cleaned and prepared for conservation. It's well-known that the animals inside the shells can influence their ability to resist heat stress. Moreover, luminance can be significantly affected by the environment in which it is studied. The authors claim their preliminary experiment to check this was not significant (lines 137-145), but this might be due to insufficient statistical power. The preliminary checking used N=18, whereas the main experiment used N=180/150. A more robust experiment checking with at least N=30 per environmental treatment might yield significant results.

Prediction and Experimentation. The prediction that lighter shells will show higher luminance or brightness is obvious and adds nothing new. The critical question is whether darker live snails heat up more than lighter ones under natural conditions, and if these differences in heating capability influence individual fitness in nature. These experiments have not yet been conducted.

Reviewer #2: I am pleased to have the opportunity to review your manuscript entitled “Shell colour luminance of Cuban painted snails, Polymita picta and Polymita muscarum (Gastropoda: Cepolidae)”. In this study, the authors successfully demonstrated the shell colour measurement and showed the difference between colour morphs. I agree this study established the methodological baseline for the future study of Polymita snails. This research undoubtedly offers a substantial contribution to PLoS One. Comments which I pointed out are only a few minor points.

L53: Is this the correct way to number the references? Is it OK if it isn't in order of appearance?

L90: “we”

L123: I understood that the authors took photos with the shell apex facing up in your study. I wonder if there are snails with banding patterns that only appear on the bottom. If yes, would these banding patterns affect your result? For example, when a snail estivates on a tree, I was concerned that bands on the bottom could be seen. Also, I recommend you add the example photo to measure the luminance as the supplemental picture so that readers can easily understand how the authors took them.

L185: I think you're right, but it is better to compare the variances by doing the statistical test, such as the F-test or something?

L187: In Figure 2, the number of individuals is not written. I can read the numbers from the main text, but I think it would be kind if you added them to match the other figures.

L199: What did it mean? High density and homogenous?

L271: From the figures, it looks like P. muscarum is brighter, but did you compare luminance values using a statistical test? And, is it correct to recognise that you would like to compare between two species, including all colours from black to yellow or albino?

L274: Why did you introduce P. picta as a subspecies here?

Reviewer #3: This is a good paper reporting some interesting findings about the luminance of various colour morphs of the Cuban painted snails, Polymita picta and P. muscarum. I really enjoyed reading about this beautiful and less studied land snail, compared to e.g., C. nemoralis. I think the authors did a good job in investigating the link between colour and thermoregulation. The analyses are sound, and appropriate statistical analyses were carried out and the findings were well presented. There is a good understanding of the literature, and the authors adequately discussed the results in relation to previous literature, as well as suggesting avenues for future research. I do however have some minor reservations: Specifically, the raw dataset of measurements for each snail in both species should be made available to the reader beyond the tables of mean values provided in the manuscript. Additionally, there are some issues in the writing that needs addressing to make the text clearer and more understandable. Once that is completed, I am happy to suggest that only minor revisions are necessary before this manuscript is suitable for publication.

See below for specific minor issues that should be addressed.

Line 31: Remove the full stop between “using digital tools.” And “We were able”.

Line 39-41: The sentence is unclear, consider revising to something along the lines of: “Luminance differences support the hypothesis that colour possesses an adaptive value for thermoregulation, not only for the background colour, but also for the colour of the band, suggesting a complex pigment composition.”

Line 61:62: Recognised used twice with both American and British spelling. Consider revising to “-recognized indicators for climate change – are known threats for land snails”.

Line 70: Should be “has predominantly focused on non-Neotropical gastropods”

Line 88-90: It is not clear what “earlier methodology” the authors are referring to here. Consider specifying and elaborate on this in the text.

Line 90: “wee” should be “we”.

Line 131: The term “till the borders” makes little sense. Is it supposed to be “to the borders”?

Line 133: Is Histogram a function within a software or a standalone software? Would be good to specify.

Line 141: Consider revising to “Three conditions for imaging were tested”.

Line 150: “in three morphological types” should be “into three morphological types”.

Line 176: Species name should be in italics, the same is true for all species names in sub-headers.

Line 194: Consider revising to “The highest luminance values”

Line 199: “The black morph had the highest luminance values around the statistical mean” This statement is confusing: Firstly, It would be nice to state what the statistical mean is. Secondly, from looking at figure 3 and reading the text this might be a misstatement? The text prior to this statement clearly states that the lowest values were observed in black shells, which is also apparent from Figure 3. Consider revising to make this statement clearer and correspond to figure 3.

Line 235-238: The sentence flows strangely and is hard to understand, consider revising (e.g., add a comma after ‘albedo’)

Line 258: “According to us” consider revising to “According to the results presented here”

Line 267: Should read “measurements of oxidative stress”

Line 278: I don’t understand what the statement “mating longer” means. Please clarify.

Line 289: Consider revising to “This study shows that using a more solid statistical approach” for clarity.

Line 292: The sentence “Our findings suggest evolutionary mechanisms play an important role.” Is somewhat non-sensical. Played an important role in how and in what? Yes, evolutionary mechanisms play an important role in the development of phenotypes, this sentence needs more context. Consider revising to be more relevant to the research presented. E.g., something along the lines of “Our results indicate that thermal selection is a key evolutionary mechanism driving the contemporary distinct distribution of both subspecies.

6. PLOS authors have the option to publish the peer review history of their article (what does this mean?). If published, this will include your full peer review and any attached files.

Reviewer #1: **Yes: **Emilio Rolán-Alvarez

Reviewer #2: No

Reviewer #3: No

---

## [Author Response · Author response to Decision Letter 0]

31 Oct 2024

October 27th , 2024

Dear Reviewers,

We deeply appreciate the thoroughness of your review and each of your questions and recommendations. Without a doubt, they have been crucial in refining our experimental design and enriching both our theoretical and methodological framework, as well as our results and discussion. Below, we address your comments.

Reviewer #1

We understand that your first critical observation refers to the Schilk (1975) methodology for calculating luminance, using the equation Y = 0.299R + 0.587G + 0.114B, where Y is the luminance, and R, G, and B represent the red, green, and blue channels, respectively. More modern formulas can be found as recommendations for calculating luminance in digital images. For example, the International Telecommunication Union (ITU) and the International Commission on Illumination (CIE) recommend the equation Y = 0.2126R + 0.7152G + 0.0722B (Wyszecki & Stiles, 2000; ITU-R BT.709-6, 2015), which is calibrated to human vision and applied in audiovisual technologies. However, Schilk’s (1975) proposal uses coefficients different from those of ITU and CIE, adjusting the parameters to spectral reflectance measurements of mollusk shells, taking into account the material properties. For this reason, we believe it is the most suitable formula for our study. In addition, this formula (or a similar approach) has been previously used to assess the shell colour luminance of other land snail species (Chiba, 1999; Ito and Konuma, 2020; Yamagishi et al., 2020; Ito et al., 2021).

Following this, you comment in your review:

"The manuscript, however, lacks substantial scientific value, because it studies something obvious which does not add anything new to the system (see below) and uses very little experimental effort (see below). The authors should revise their objectives and methodology to meet international standards before attempting publication."

It is true that the introduction previously mentioned two study goals. We have unified these as follows: "In this study, we aim to characterize the shell colour luminance in Polymita picta and Polymita muscarum, as well as explore its potential relationship with thermoregulation" (lines 118–119). Furthermore, the earlier goal, "Our study aims to better understand the variation in shell luminance among different colour morphs and band patterns in P. muscarum and P. picta to subsequently explore their potential role in thermoresistance," was developed by quantifying shell luminance using digital images. In our view, this study establishes the methodological basis for future research on Polymita luminance and other species in Cuba by exploring lighting conditions, and it characterizes this colour parameter in the different morphological groups included in the investigation. The values obtained and the analysis of their variation represent the first quantification of colour and a baseline in the Polymita genus—an attractive taxa, particularly as a potential Caribbean model to explore multiple pathways of the evolutionary forces in modulating the rich diversity of traits in living forms.

Next, in your review, you state:

"Literature Review. Many classical references on Polymita (in Spanish) are of low scientific quality or are supposed to prove what they actually do not. It might be beneficial to prepare an English review that objectively summarizes previous work and realistically presents the data. Currently, there is no clear proof that Polymita polymorphism is maintained by natural selection, despite some indirect suggestions. A comprehensive review could set a solid foundation for future research."

Unfortunately, most publications on Polymita colour, especially regarding P. picta roseolimbata and P. muscarum and, to a lesser extent, P. venusta, were conducted between the 1980s and the early 2000s. Mostly, these articles were published in Spanish, mainly in different Cuban journals. However, by first time, these publications summarized and quantified key aspects of the ecology, genetic polymorphism, and reproduction of Polymita species, making them essential references.

In 1992, Berovides and Alfonso published experimental evidence of climatic selection in the subspecies Polymita picta roseolimbata. They exposed different morphs to sunlight and infrared light, identifying those most resistant to dry and humid heat. Other studies on the same subspecies also identify segregation of some morphs among different vegetation formations, but in all cases, the base colour and band traits were treated as "discrete." Today, we know that within these distinguishable colour ranges, there is continuity, with morphologically and genetically differentiable phenotypes, as shown through quantitative colour analysis techniques in species like Cepaea nemoralis (Davison et al., 2024).

We have addressed the need for a more in-depth discussion of the evolutionary aspects involved in Polymita shell colour polymorphism, and this recommendation has been particularly valuable for our introduction (lines 90–93, 96–106). This new approach also allows us to better explain novel results, such as the presence of distinguishable groups within a single morph with potential ecological and evolutionary relevance, identifiable through luminance quantification (lines 287–296). Likewise, we have expanded the discussion regarding the action of evolutive mechanisms in colour polymorphism in both Polymita species and the potential presence of other related genetic processes (lines 297–306).

Further in your review, you mention:

"Use of Luminance. The methodology for using luminance as presented has little empirical value, especially with museum specimens that have been cleaned and prepared for conservation. It's well-known that the animals inside the shells can influence their ability to resist heat stress. Moreover, luminance can be significantly affected by the environment in which it is studied."

Indeed, luminance values can be affected by various environmental factors such as light intensity, the refractive index of the medium, or turbulence, as well as by individual factors like the animal’s soft body colour. The variability in Polymita tegument colour ranges from light gray to black (Torre, 1950; González-Guillén, 2014). The study of empty shells does not account for variability introduced by the animal's colour and physiological state, although these variables undoubtedly play a crucial role in thermoregulation, as do factors like shell size and opening, porosity, and adaptations of the chaperone system, among others. However, the use of empty shells to analyse their optical properties is methodologically accepted (e.g., Savazzi & Sasaki, 2013). For this study, we selected the most recent shells whenever possible and ensured they showed no appreciable signs of erosion or discolouration.

You further note:

"The authors claim their preliminary experiment to check this was not significant (lines 137–145), but this might be due to insufficient statistical power. The preliminary checking used N=18, whereas the main experiment used N=180/150. A more robust experiment checking with at least N=30 per environmental treatment might yield significant results."

In the previously submitted version, we stated:

"In four of five phenotypic groups, no significant differences between the mean shell colour luminance for the three light conditions were detected (Fig 1 A-E). Only for the unbanded brown shells of P. muscarum (Fig 1 B), the luminance mean was significantly different under artificial light (H = 11.6, p = 0.003). Therefore, we decided to proceed with all subsequent analyses under artificial light conditions."

Considering your pertinent observation regarding sample size, we refined our statistical processing to increase resolution. A repeated-measures ANOVA was used for groups meeting normal distribution and homoscedasticity criteria, while the Friedman test was applied in the case of non-normal distribution. In the group where significant differences were found, we performed a Wilcoxon test with Bonferroni correction to increase sensitivity, followed by an eta-squared test to evaluate the effect size of the independent variable on luminance variation. As in the previous manuscript version, we decided to continue experiments under artificial light, mentioning the sample size limitations (lines 177–181, 189–199).

Lastly, you mention:

"Prediction and Experimentation. The prediction that lighter shells will show higher luminance or brightness is obvious and adds nothing new. The critical question is whether darker live snails heat up more than lighter ones under natural conditions, and if these differences in heating capability influence individual fitness in nature. These experiments have not yet been conducted."

Certainly, further experiments are needed to better understand the relationship between Polymita shell colour polymorphism, luminance values, thermoregulation, and fitness in nature. Nevertheless, we have previously mentioned the novel contributions of this study, which has been enriched by your various suggestions (lines 119–123, 356–362, 365–368, 373–376).

Reviewer #2

We have carefully considered your suggestions and reordered the references in the text according to the PLOS One guidelines. Additionally, we have addressed all of your grammatical and writing recommendations and has have added the sample size for each group to Figure 2.

We proceed by addressing your questions below.

"L123: I understood that the authors took photos with the shell apex facing up in your study. I wonder if there are snails with banding patterns that only appear on the bottom. If yes, would these banding patterns affect your result? For example, when a snail estivates on a tree, I was concerned that bands on the bottom could be seen. Also, I recommend you add the example photo to measure the luminance as the supplemental picture so that readers can easily understand how the authors took them."

We have outlined the criteria considered for using the dorsal view of the shells in the luminance calculation (lines 140–147) and have also provided as supplementary material the luminance data for both dorsal and ventral views of 180 individuals (S1 Table). No significant differences in luminance between the photographs of the two views were detected.

"L185: I think you're right, but it is better to compare the variances by doing the statistical test, such as the F-test or something?"

Prior to other statistical analyses, all data matrices were assessed for normality and homoscedasticity (lines 176–177), and in most cases, variance homogeneity was detected. Consequently, differences in central values, rather than dispersion, were explored.

"L199: What did it mean? High density and homogenous?"

We understand that this section of the manuscript was not written with complete clarity. We refer to the dispersion of the data observed in the probability density diagrams in the violin plots. In the new version, we aimed to clarify the wording of these results and their discussion (lines 213–216, 229–232, 287–296).

"L271: From the figures, it looks like P. muscarum is brighter, but did you compare luminance values using a statistical test? And, is it correct to recognize that you would like to compare between two species, including all colours from black to yellow or albino?"

You are correct that methodologically it is not appropriate to compare both species using direct statistical procedures. However, it is indeed insightful to comment on the graphical and mean luminance differences between the two species, as these differences likely reflect variations in their life histories. In future research, a more controlled comparison, possibly including a larger sample size across all morphs and colour ranges (from black to yellow or albino), would be highly beneficial to better understand these patterns.

"L274: Why did you introduce P. picta as a subspecies here?"

The genus Polymita consists of two subgenera, following the description by Torre (1950): the subgenus Polymita and the subgenus Oligomita. Regarding Polymita picta, five subspecies have been described: P. picta picta, P. picta iolimbata, P. picta fuscolimbata, P. picta nigrolimbata, and P. picta roseolimbata. Most studies on genetic polymorphism and ecology of this species have focused on the subspecies P. picta roseolimbata and P. picta nigrolimbata. However, as mentioned in the manuscript, these subspecies require further investigation through molecular genetics and new studies on their biogeography and behaviour, which could offer a more precise taxonomic revision.

Reviewer #3

We have addressed each of your grammatical and writing recommendations. Furthermore, we have provided five supplementary materials, making available all the data used in our statistical analyses and figure construction. 

Sincerely,

Mario J. Gordillo-Pérez

Centre for Environmental Sciences, Hasselt University, Belgium

+32 497750324

References

1. Schlick, C. (1995). Quantization techniques for visualization of high dynamic range pictures. In G. Sakas, P. Shirley, & S. Muller (Eds.), Photorealistic rendering techniques (pp. 7-20). Springer.

2. International Telecommunication Union. (2015). ITU-R Recommendation BT.709-6: Parameter values for the HDTV standards for production and international programme exchange. International Telecommunication Union.

3. Wyszecki, G., & Stiles, W. S. (2000). Color science: Concepts and methods, quantitative data and formulae (2nd ed.). Wiley-Interscience.

4. Chiba S. Character displacement, frequency-dependent selection, and divergence of shell colour in land snails Mandarina (Pulmonata). Biological Journal of the Linnean Society. 1999; 66: 463–479.

5. Ito S, Konuma J. Disruptive selection of shell colour in land snails: a mark–recapture study of Euhadra peliomphala simodae. Biological Journal of the Linnean Society. 2020; 129(2): 323–333. https://doi.org/10.1093/biolinnean/blz168.

6. Yamagishi M, Ito S, Konuma J. Record of an albino land snail Euhadra quaesita. American Malacological Bulletin. 2020; 38(1): 1–3.

7. Ito S, Hirano T, Chiba S, Konuma J. Shell colour diversification induced by ecological release: a shift in natural selection after a migration event. Ecology and Evolution. 2021; 11: 15534–15544. https://doi.org/10.1002/ece3.8080.

8. Berovides, V., & Alfonso, M. A. (1992). Evidencias experimentales de la selección climática en Polymita picta roseolimbata (Gastropoda: Fruticicollidae) de Maisí, Cuba. Ciencias Biológicas, 25, 1-8.

9. Torre, C. (1950). El género Polymita. Memorias de la Sociedad Cubana de Historia Natural “Felipe Poey", 20, 1-20.

10. González-Guillén, A. (2014). Polymita: The most beautiful land snail of the world (1st ed.). Fundcraft Publishing.

11. Savazzi, E., & Sasaki, T. (2013). Observations on land-snail shells in near-ultraviolet, visible and near-infrared radiation. Journal of Molluscan Studies, 79, 95-111.

---

## [Editor Report · Decision Letter 1]

5 Nov 2024

Shell colour luminance of Cuban painted snails, Polymita picta and Polymita muscarum (Gastropoda: Cepolidae)

PONE-D-24-25732R1

Dear Dr. Gordillo Pérez,

We’re pleased to inform you that your manuscript has been judged scientifically suitable for publication and will be formally accepted for publication once it meets all outstanding technical requirements.

Kind regards,

Hudson Alves Pinto, Ph.D

Academic Editor

PLOS ONE

Additional Editor Comments (optional):

I thank the Authors for their efforts and elegance in revising this MS. The text improved a lot and is now suitable for publication. Congratulations on this nice work.

Reviewers' comments:

None

---

## [Editor Report · Acceptance letter]

11 Nov 2024

PONE-D-24-25732R1 

PLOS ONE

Dear Dr. Gordillo Pérez, 

I'm pleased to inform you that your manuscript has been deemed suitable for publication in PLOS ONE. Congratulations! Your manuscript is now being handed over to our production team.

Kind regards, 

on behalf of

Dr. Hudson Alves Pinto 

Academic Editor

PLOS ONE